

# Dose-response effect of L-citrulline on skeletal muscle damage after acute eccentric exercise: an *in vivo* study in mice

Dhoni Akbar Ghozali[1,2], Muchsin Doewes[2], Soetrisno Soetrisno[3], Dono Indarto[4] and Muhana Fawwazy Ilyas[1,5]

[1] Department of Anatomy and Embryology, Universitas Sebelas Maret, Surakarta, Central Java, Indonesia
[2] Doctoral Program of Medical Sciences, Faculty of Medicine, Universitas Sebelas Maret, Surakarta, Central Java, Indonesia
[3] Departement of Obstetrics and Gynecology, Faculty of Medicine, Universitas Sebelas Maret, Surakarta, Central Java, Indonesia
[4] Department of Physiology, Faculty of Medicine, Universitas Sebelas Maret, Surakarta, Central Java, Indonesia
[5] Department of Neurology, Universitas Sebelas Maret, Surakarta, Central Java, Indonesia

Corresponding author
Dhoni Akbar Ghozali,
dhoniakbar@staff.uns.ac.id

## ABSTRACT

**Background**. Eccentric exercise may trigger mechanical stress, resulting in muscle damage that may decrease athletic performance. L-citrulline potentially prevents skeletal muscle damage after acute eccentric exercise. This study aimed to assess the dose-response effect of L-citrulline as a preventive therapy for skeletal muscle damage in mice after acute eccentric exercise.

**Methods**. This is a controlled laboratory *in vivo* study with a post-test-only design. Male mice (BALB/c, $n = 25$) were randomized into the following groups: a normal control (C1) ($n = 5$); a negative control (C2) with downhill running and placebo intervention ($n = 5$); treatment groups: T1 ($n = 5$), T2 ($n = 5$), and T3 ($n = 5$), were subjected to downhill running and 250, 500, and 1,000 mg/kg of L-citrulline, respectively, for seven days. Blood plasma was used to determine the levels of TNNI2 and gastrocnemius muscle tissue NOX2, IL-6, and caspase 3 using ELISA. NF-κB and HSP-70 expressions were determined by immunohistochemistry.

**Results**. Skeletal muscle damage (plasma TNNI2 levels) in mice after eccentric exercise was lower after 250 and 500 mg/kg of L-citrulline. Further, changes in oxidative stress markers, NOX2, were reduced after a 1,000 mg/kg dose. However, a lower level of change has been observed in levels of cellular response markers (NF-κB, HSP-70, IL-6, and caspase 3) after administration of L-citrulline doses of 250, 500, and 1,000 mg/kg.

**Conclusion**. L-citrulline may prevent skeletal muscle damage in mice after acute eccentric exercise through antioxidant effects as well as inflammatory and apoptotic pathways. In relation to dose-related effects, it was found that L-citrulline doses of 250, 500, and 1,000 mg/kg significantly influenced the expression of NF-κB and HSP-70, as well as the levels of IL-6 and caspase 3. Meanwhile, only doses of 250 and 500 mg/kg had an impact on TNNI2 levels, and the 1,000 mg/kg dose affected NOX2 levels.

## INTRODUCTION

Muscle injuries are very common in sports, with 10–55% incidence (*Ingham et al., 2017*). The most common injuries are strains caused by excessive muscle tension. High-intensity exercises, particularly eccentric exercises, often cause mild-level strain (*Veqar & Kalra, 2013*). Eccentric exercise may trigger mechanical stress, resulting in muscle damage. Muscle damage requires a long recovery time and restricts athletes from performing their best (*Yan et al., 2016*). Hence, significant research is going on to discover more effective and efficient approaches to enhance the performances of athletes with muscle damage. One common and popular method of maximizing performance is to use ergogenic supplements (*Cribb & Hayes, 2006*). One of the ergogenic supplements that may have a preventive effect on skeletal muscle damage is L-citrulline (*Harty et al., 2019*; *Gonzalez & Trexler, 2020*; *Aguayo et al., 2021*). Preliminary review shows that amino acids, one of which is L-citrulline *via* their regulatory properties, may directly influence the proteome (*Bourgoin-Voillard et al., 2016*). L-citrulline is a non-essential amino acid found primarily in watermelon (*Citrullus vulgaris*). L-citrulline is an endogenous precursor of L-arginine, a substrate for nitric oxide synthase (NOS). L-citrulline is successfully recycled through the nitric oxide (NO) cycle to L-arginine and plays a crucial role in NO metabolism and regulation (*Bescós et al., 2012*). L-citrulline is known that has a modulatory effect on the inflammatory response and is more effective than L-arginine. Further, L-citrulline can reduce the response of pro-inflammatory mediators (such as IL-6) without interfering with the secretion of anti-inflammatory mediators (such as IL-10) (*Asgeirsson et al., 2011*). L-citrulline also showed to decrease reactive oxygen species (ROS) production by reducing the expression of p67$^{phox}$, which is the main component of nicotinamide adenine dinucleotide phosphate (NADPH) oxidase 2 (NOX2) (*Tsuboi, Maeda & Hayashi, 2018*).

Several studies have been conducted, some of the results are L-citrulline (250 mg/kg) for seven days effectively increased exercise performance in mice (*Takeda et al., 2011*). Besides that, watermelon juice (1.2 g L-citrulline) and fortified watermelon juice (6 g L-citrulline) have been reported to reduce muscle soreness after 24 h of exercise (*Tarazona-Díaz et al., 2013*). L-citrulline supplementation (5 g/kg/day) for one week also showed the capability of modulating muscle function by showing improvement of muscle mass and motor activity, which was highly associated with that of maximal tetanic isometric force (*Faure et al., 2012*). Furthermore, L-citrulline supplementation (5 g/kg/day) also led to higher protein synthesis and protein content in muscle (*Osowska et al., 2006*). In addition, with different dose, L-citrulline (1.80 g/kg) has actions on muscle protein synthesis (MPS) with the capacity to stimulate the mechanistic target of rapamycin complex 1 (mTORC1) pathway (*Le Plénier et al., 2012*).

To date, studies assessing the antioxidant, anti-inflammatory, and anti-apoptotic effects of L-citrulline on skeletal muscle damage after acute eccentric exercises are limited. Therefore, this study aimed to investigate the dose–response effect of L-citrulline on skeletal muscle damage by measuring the levels of NOX2, IL-6, caspase 3, and TNNI2 and expression of NF-κB and HSP-70 in mice after an acute eccentric exercise. TNNI2 was proposed because it is a sensitive and fast fiber-specific serum marker of skeletal

muscle injury (*Simpson et al., 2005*; *Chapman et al., 2013a*; *De Matteis et al., 2019*). It is still rarely for being utilized, so it may give more valuable insight compared to common marker including creatine kinase (CK). Furthermore, the positioning of HSP-70 both upstream and downstream in the stress-induced apoptosis pathway suggests a mechanism for ensuring death that can be inhibited (*Park et al., 2017*), but there has been no further investigation in the effect of L-citrulline this area. In this study, ELISA was used due to the likelihood that TNNI2 primarily circulates in the bloodstream, as well as NOX2, IL-6 (myokine), and caspase 3 in muscle tissue. In contrast, NF-κB and HSP 70 may be more localized to specific tissues, which is why immunohistochemistry was utilized.

## MATERIALS & METHODS

### Study design

This was a controlled laboratory *in vivo* study with a post-test-only design. This study was conducted from October 2022 to January 2023. The sample size was calculated using ANOVA design: degrees of freedom divided by the sum of the groups plus one (*Wan Mohammad, 2017*). This calculation demonstrated a sample size of 25. *Mus musculus* Balb/c mice (8 weeks old and weighing 25.79–28.37 g) were obtained from the Experimental Animal Laboratory, Department of Physiology and Medical Biochemistry, Faculty of Medicine, Universitas Airlangga, Indonesia. Animals with any deformity, injury, or inflammation of the forelimbs or hind limbs were excluded. This study was approved by The Research Ethics Committee of the Faculty of Medicine, Universitas Sebelas Maret, Indonesia, with protocol number 01/02/09/2022/117. This study conformed with Animal Research: Reporting of In Vivo Experiments (ARRIVE) guidelines. A total of 25 mice were randomly divided into five groups: two control groups (normal/C1 and negative/C2) and three treatment groups (T1, T2, and T3), with each group contains five mice. A laboratory assistant performed the randomization process, and the authors were blinded for each group.

### Animal care, feeding, housing, and enrichment

Mice were placed in cages (37 cm × 28 cm × 13 cm) covered with a wire on the top so they could move freely and were not stressed. The mice were acclimatized for one week under reversed light-dark conditions (12 h light and 12 h dark) at 23–26 °C and 40–60% humidity. The mice received a standard feed (15% BW/day) and drinking water *ad libitum*.

### Treatment procedure

Once acclimatization was completed, mice in the C2 group received tap water, whereas mice in the T1, T2, and T3 groups were administered 250, 500, and 1,000 mg/kg of BW/day L-citrulline (C7629; Sigma-Aldrich, St. Louis, MO, USA) for seven days, respectively. The administration of L-citrulline was performed through oral gavage inserted into the esophagus to ensure precise dosing. In day seven, four hours after the absorbance, all mice, except the C1 group, performed downhill exercises using the protocol described previously (*Purwanto, Harjanto & Sudiana, 2016*). The mice were allowed to adapt to the Colombus Treadmill (Columbus Instruments, Columbus, OH, USA) for 5 min, and subsequently,

downhill running was conducted at a speed of 30 cm/s for 18 min at a $-15°$ declination angle with a single running frequency.

## Euthanasia and post-study procedures

At the end of the experiment, surviving animals were euthanized using $CO_2$ asphyxiation following American Veterinary Medical Association (AVMA) guidelines. It is the most common and acceptable method of euthanasia for mice (*Underwood & Anthony, 2020*). It reliably and rapidly induces loss of consciousness with minimal distress (*Makowska et al., 2009*; *Boivin et al., 2017*).

## Data retrieval

After four hours of exercise, mice were anesthetized using an intraperitoneal injection of 5 mg/25 g BW ketamine, and venous blood samples were collected from intracardiac aspiration (preferred for consistency) using a 3 ml syringe. Subsequently, blood samples were centrifuged at 3,000 rpm for 20 min, and collected plasma samples were stored at $-80°C$ before further analysis. The mice's right and left gastrocnemius muscles were prepared using scissors, scalpels, and tweezers by cutting both sides of the origin and insertion tendons. The right gastrocnemius muscle was transferred to a tissue grinder tube filled with cold phosphate-buffered saline (PBS) (Thermo Fisher Scientific, Waltham, MA, USA). In contrast, the left gastrocnemius muscle was embedded in 10% buffered formalin (Fisher Chemical, Hampton, NH, USA) for immunohistochemical staining.

## Measurements of NOX2, IL-6, caspase 3, and TNNI2 levels

Muscle NOX2, IL-6, and caspase 3 levels and plasma TNNI2 levels were examined at the Biomedical Laboratory, Universitas Sebelas Maret, Indonesia, using Mouse NOX2 (BZ-22142589-EB; BIOENZY, Jakarta, Indonesia), IL-6 (BZ-08149400-EB; BIOENZY, Jakarta, Indonesia), and caspase 3 (BZ-08143151-EB; BIOENZY, Jakarta, Indonesia) ELISA kits and mouse Troponin I, Fast Skeletal Muscle (TNNI2) ELISA kit (MBS927961; MyBioSource, San Diego, CA, USA), respectively, according to manufacturer's instructions. For the assessment, reagents, standards, controls, and samples were prepared at room temperature. A 100 mg sample was weighed, homogenized using a sterile mortar, and transferred to a 1.5 mL sterile microtube. Subsequently, 500 μL of lysis buffer was added, and the sample was sonicated, followed by centrifugation at 14,000 rpm at 4 °C for 20 min. The NOX2 standard was diluted (dilution factor of 6), and both the standard (50 μL) and sample aliquots (40 μL) were added to the well plate. Biotin conjugate (10 μL) and horseradish peroxidase (HRP) (50 μL) were introduced into all well plates. Incubation was carried out at 37 °C for 90 min, followed by repeated washing with 350 μL of washing buffer five times. Detection Reagent A (50 μL) and Detection Reagent B (50 μL) were added to the well plate, which was then sealed and incubated at 37 °C for 30 min. The reaction was terminated by adding 50 μL of Stop Solution, and the absorbance was measured at a wavelength of 450 nm. The procedure was consistently employed to assess the levels of myokine (IL6) with a dilution factor of 6, caspase 3 with a dilution factor of 7, and TNNI2 with a dilution factor of 7, utilizing customized reagents and standards tailored to each specific biomarker. Two experts performed the blinded examination.

## Measurements of NF-κB and HSP-70 expression

Expression of NF-κB and HSP-70 was determined at the Anatomical Pathology Laboratory, Universitas Sebelas Maret, Indonesia, using immunohistochemical staining using NF-κB p65 (F-6 sc-8,008; Santa Cruz Biotechnology Inc., Santa Cruz, CA, USA) and anti-HSP-70 (sc-24; Santa Cruz Biotechnology Inc., Santa Cruz, CA, USA) antibodies using manufactures' instructions. First, paraffin-embedded tissue blocks, sliced to a thickness of 4–5 μm, were placed on poly-L-lysine-coated slides and incubated overnight at 37 °C to ensure adherence. Subsequently, the samples underwent deparaffinization, which included a series of xylene and alcohol treatments to remove paraffin and prepare the tissue for staining. After that, antigen retrieval was carried out in a microwave oven using Tris EDTA buffer at pH 9, with subsequent cooling and further rinsing. The samples were then treated to block endogenous peroxidase activity with 3% hydrogen peroxide in methanol, followed by a serum blocking step. Next, the tissue sections were incubated with prepared antibodies specific to NF-κB or HSP 70, maintained at 4 °C for 18 h to allow for antibody binding. Following antibody incubation, the samples underwent additional washes with PBS, and a biotin solution was applied for 15 min, followed by streptavidin for 10 min. To visualize the target proteins, a peroxidase enzyme substrate (DAB) was added and left for 3–5 min. After staining, the samples were thoroughly washed with running water and counterstained with hematoxylin for 4 min to provide contrast to the stained areas. Finally, the tissue sections were mounted and covered with a coverslip for examination. Blinded readings were obtained by two experts using an Olympus CX23 light microscope (Olympus Corporation, Tokyo, Japan) at 40 times magnification. The tests were judged positive and negative based on the staining results of muscle cells (myocytes); brown and blue staining was considered positive and negative, respectively. The percentage of cells showing positive staining was calculated by dividing the number of positive cells by the number of all muscle cells.

## Statistical analysis

Descriptive analysis was used to determine the data distribution and concentration. The Shapiro–Wilk test was performed to determine the distribution of the data. Levene's test was used to determine the homogeneity of the data between groups. ANOVA and Fisher's LSD posthoc tests were performed to analyze the differences in each variable among the five study groups. All statistical tests were two-sided, and $P$-values of $< .05$ were considered statistically significant. Statistical analyses were performed using IBM SPSS Statistics for Windows (version 21.0; IBM Corp. Armonk, NY, USA).

## RESULTS

Twenty-five mice ($n = 5$, per each group) enrolled in the study (none of the mice had adverse events and were excluded from the analysis). Table 1 shows the TNNI2, NOX2, caspase 3, IL-6, NF-κB, and HSP-70 levels in entire study groups. Meanwhile, the differences for each group can be seen in Fig. 1. Furthermore, the immunochemistry results for NF-κB and HSP-70 expressions can be seen in Fig. 2.

**Table 1  Levels of NOX2, IL-6, caspase 3, TNNI2, NF-κB, and HSP-70.**

| Group | NOX2, ng/mL | NF-κB, % | HSP 70, % | IL-6, ng/mL | caspase 3, ng/mL | TNNI2, pg/mL |
|---|---|---|---|---|---|---|
| C1 ($n = 5$) | 2,33 ± 0,70 | 39,00 ± 10,25 | 46,00 ± 10,84 | 8,25 ± 1,09 | 1,67 ± 0,13 | 6,41 ± 7,67 |
| C2 ($n = 5$) | 3,66 ± 1,42 | 75,00 ± 11,18 | 78,00 ± 8,37 | 15,31 ± 2,02 | 2,73 ± 0,47 | 350,77 ± 246,80 |
| T1 ($n = 5$) | 3,84 ± 0,51 | 40,00 ± 10,00 | 48,00 ± 7,58 | 8,55 ± 0,69 | 1,72 ± 0,19 | 48,00 ± 40,53 |
| T2 ($n = 5$) | 3,84 ± 0,80 | 41,00 ± 5,48 | 51,00 ± 11,40 | 9,35 ± 0,61 | 1,87 ± 0,38 | 176,21 ± 76,61 |
| T3 ($n = 5$) | 2,12 ± 0,78 | 42,00 ± 8,37 | 52,00 ± 12,55 | 8,55 ± 1,13 | 2,16 ± 0,32 | 195,78 ± 42,88 |
| $P$-value | **.002** | **<.001** | **<.001** | **.002** | **.002** | **.009** |

Notes.

Data were presented as mean ± standard deviation. $P$-values (indicated in bold) were calculated using the analysis of variance.

*$P$-value of < .05 indicates statistical significance.

NOX2, NADPH oxidase 2; HSP-70, heat shock protein-70; IL-6, interleukin 6; TNNI2, Fast Skeletal Muscle Troponin I; C1, placebo only; C2, placebo with downhill running protocol; T1, L-citrulline supplement 250 mg/kg body weight for 7 days with downhill running protocol; T2, L-citrulline supplement 500 mg/kg body weight for 7 days with downhill running protocol; T3, L-citrulline supplement 1,000 mg/kg body weight for 7 days with downhill running protocol.

### Differences in NOX2 levels

In this study, NOX2 levels in the gastrocnemius muscle tissue of mice serve as a marker of oxidative stress. NOX2 levels in the muscle tissue of mice in the T3 group (2.12 ± 0.78 ng/ml) were significantly ($P = .013$) lower than those in the C2 group (3.66 ± 1.42 ng/ml). Although NOX2 levels in the muscle tissue of mice in the T3 group were also lower than those in the C1 group (2.33 ± 0.70 ng/ml), the difference was insignificant. NOX2 levels in the T2 (3.84 ± 0.80 ng/ml, $P = 0.015$) and T3 (2.12 ± 0.78 ng/ml, $P = 0.015$) groups were significantly higher than those in the C1 group, but non-significantly higher than those in the C2 group. NOX2 levels in the T3 group were significantly ($P = 0.007$ and $P = 0.007$, respectively) lower than those in the T1 and T2 groups; however, no significant difference was observed in the NOX2 levels between the T1 and T2 groups.

### Differences in NF-κB expression

Expression levels of NF-κB in the gastrocnemius muscle tissue of mice serve as an inflammatory marker and cellular response. The expression of NF-κB in the muscle tissues of T1 (40.00 ± 10.00%, $P < .001$), T2 (41.00 ± 5.48%, $P < .001$), T3 (42.00 ± 8.37%, $P < 0.001$) groups was significantly lower than that of the C2 group (75.00 ± 11.18%) and non-significantly higher than that of the C1 group (39.00 ± 10.25%). No significant differences were observed in the levels of NF-κB among the T1, T2, and T3 groups.

### Differences in HSP-70 expression

Expression levels of HSP-70 in mice gastrocnemius muscle tissue are one of apoptosis marker and cellular responses. The expression of HSP-70 in the muscle tissues of T1 (48.00 ± 7.58%, $P < .001$), T2 (51.00 ± 11.40%, $P < 0.001$), and T3 (52.00 ± 12.55%, $P < 0.001$) groups was significantly lower than that of the C2 group (78.00 ± 8.37%) and non-significantly higher than that of the C1 group (46.00 ± 10.84%). No significant differences were observed in the levels of HSP-70 among the T1, T2, and T3 groups.

### Differences in IL-6 levels

IL-6 levels in the gastrocnemius muscle tissues of mice serve as a marker of inflammation and cellular response. The IL-6 levels in the muscle tissues of T1 (8.55 ± 0.69 ng/mL,

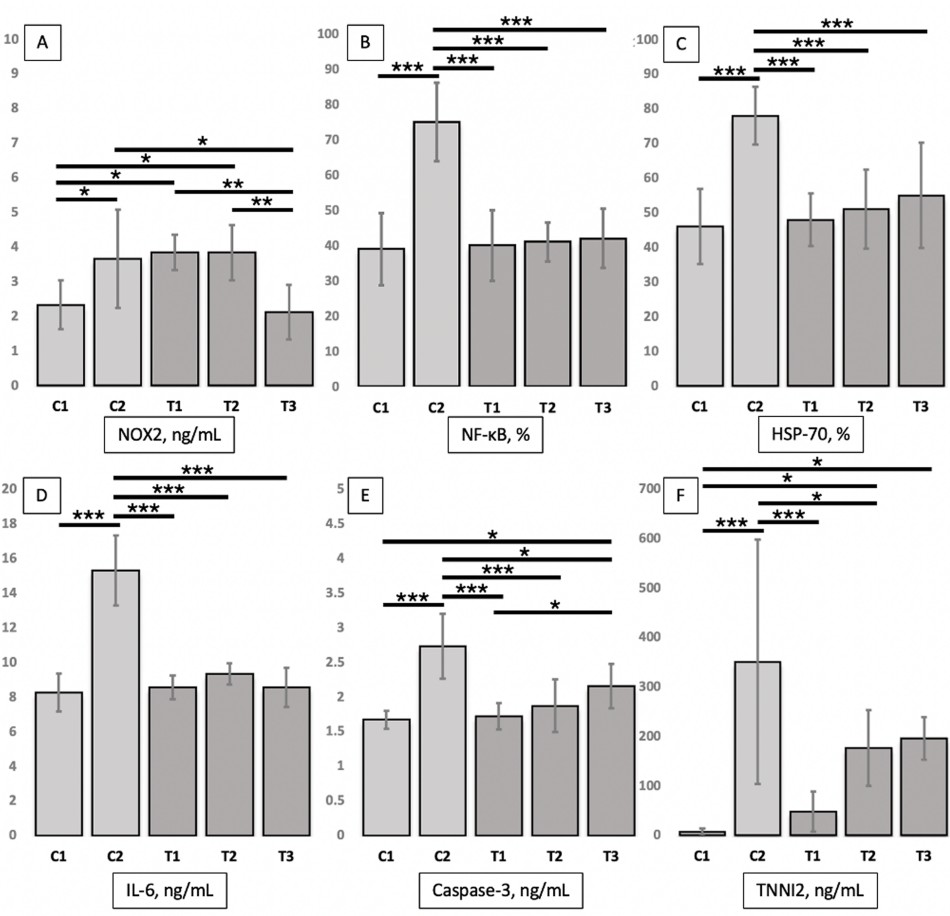

**Figure 1** (A–F) Effects of treatment (L-citrulline 250, 500, and 1,000 mg/kg) on NOX2 levels, NF-κB and HSP-70 expressions, as well as IL-6, caspase 3, and TNNI2 levels in post-eccentric exercise. Data are represented as mean ± standard deviation. Lines indicate significant differences between groups (* $P < .05$; ** $P < .01$; and *** $P < .005$). NOX2, NADPH oxidase 2; HSP-70, heat shock protein-70; IL-6, interleukin 6; TNNI2, Fast Skeletal Muscle Troponin I; C1, placebo only; C2, placebo with downhill running protocol; T1, L-citrulline supplement 250 mg/kg body weight for 7 days with downhill running protocol; T2, L-citrulline supplement 500 mg/kg body weight for 7 days with downhill running protocol; T3, L-citrulline supplement 1,000 mg/kg body weight for 7 days with downhill running protocol.

$P < .001$), T2 (9.35 ± 0.61 ng/mL, $P < .001$), and T3 (8.55 ± 1.13 ng/mL, $P < .001$) groups was significantly lower than that of the C2 group (15.31 ± 2.02 ng/mL) and non-significantly higher than that of the C1 group (8.55 ± 1.13 ng/mL). No significant differences were observed in the levels IL-6 among the T1, T2, and T3 groups.

### Differences in caspase 3 levels

Caspase 3 levels in mouse muscle tissues function as a marker of apoptosis. The caspase 3 levels in the muscle tissues of T1 (1.72 ± 0.19 ng/mL, $P < .001$), T2 (1.87 ± 0.38 ng/mL, $P < .001$), and T3 (2.16 ± 0.32 ng/mL, $P = .012$) groups were significantly lower than those of the C2 group (2.73 ± 0.47 ng/mL) and non-significantly (except those of the T3 group, $P = .027$) higher than those of the C1 group (1.67 ± 0.13 ng/mL). Furthermore,

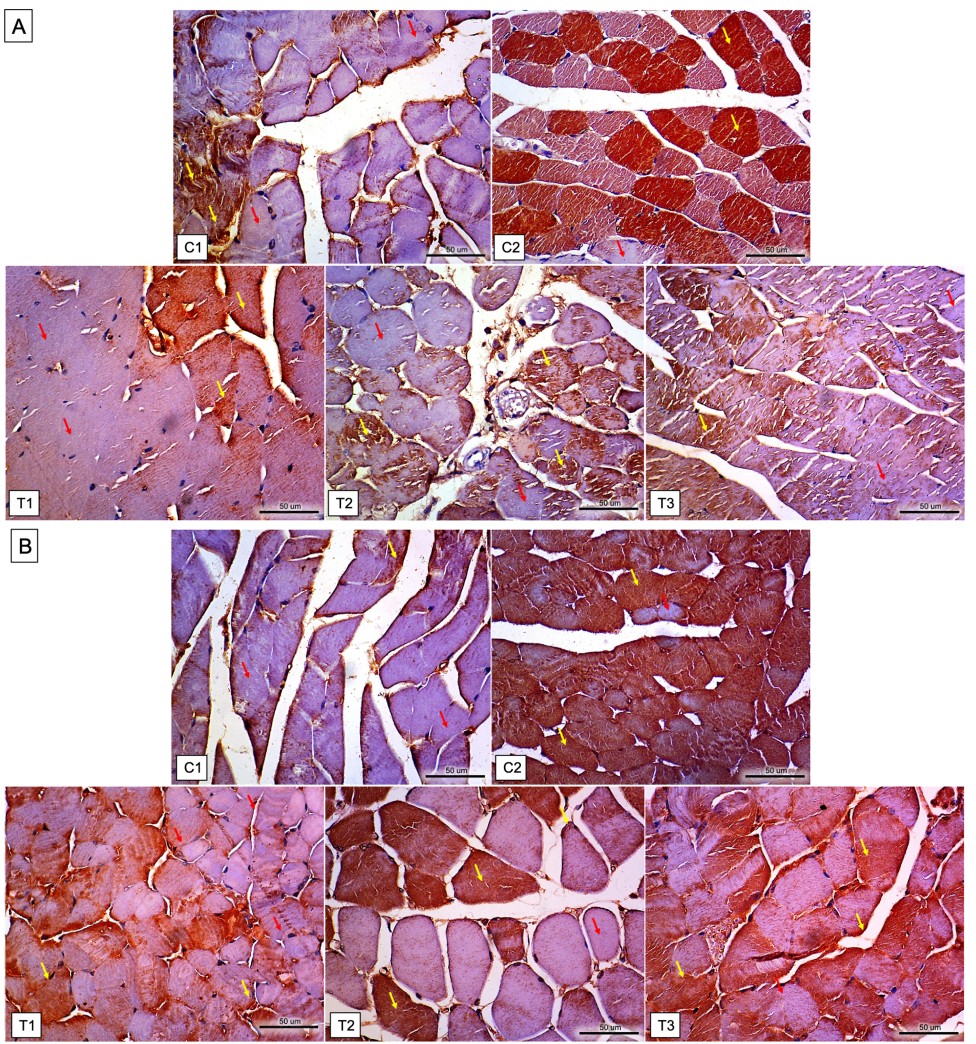

**Figure 2** **Immunohistochemistry results for NF-κB and HSP 70 expressions in the gastrocnemius muscle tissue of mice in each group.** (A) Result of NF-κB expressions: C1 = 30–50%, C2 = 60–90%, T1 = 30–50%, T2 = 35–50%, and T3 = 30–50% expression. Immunohistochemistry staining revealed the positive of NF-κB expression in the cytoplasm (indicated by the yellow arrow) and, to some extent, in the nuclei of skeletal muscle cells, whereas the expression was negative in other areas (as indicated by the red arrow). (B) Result of HSP 70 expressions: C1 = 30–55% expressions; C2 = 60–90%; T1 = 40–55%; T2 = 40–65%; and T3 = 40–65%. Immunohistochemistry staining revealed the positive of HSP 70 expression in the cytoplasm (indicated by the yellow arrow) and, to some extent, in the nuclei of skeletal muscle cells, whereas the expression was negative in other areas (as indicated by the red arrow). C1, placebo only; C2, placebo with downhill running protocol; T1, L-citrulline supplement 250 mg/kg body weight for 7 days with downhill running protocol; T2, L-citrulline supplement 500 mg/kg body weight for 7 days with downhill running protocol; T3, L-citrulline supplement 1,000 mg/kg body weight for 7 days with downhill running protocol.

the caspase 3 levels in the muscle tissue of the T1 group were significantly ($P = 0.044$) lower than those of the T3 group and non-significantly lower than those of the T2 group. No significant differences were observed in the levels of caspase 3 between the T2 and T3 groups.

### Differences in TNNI2 levels

In this study, the plasma TNNI2 levels in mice were a marker of muscle damage. Plasma TNNI2 levels of T1 ($48.00 \pm 40.53$ pg/mL, $P < .001$) and T2 ($176.21 \pm 76.61$ pg/mL, $P = .031$) groups were significantly lower than those of the C2 group ($350.77 \pm 246.80$ pg/mL) and higher (significant for T2 [$P = .035$] and T3 [$P = .020$] groups) than those of the C1 group ($6.41 \pm 7.67$ pg/mL). Plasma TNNI2 levels of the T3 group ($195.78 \pm 42.88$ pg/mL) were non-significantly lower and higher than those of the C2 and C1 groups, respectively. No significant differences were observed in the levels of TNNI2 among the T1, T2, and T3 groups.

## DISCUSSION

This study evaluated the dose–response effect of L-citrulline (250, 500, and 1,000 mg/kg) on preventing muscle damage after acute eccentric exercise. This study observed skeletal muscle damage using NOX2, NF-κB, HSP-70, IL-6, caspase 3, and TNNI2 levels. Eccentric contractions may trigger mechanical stress, which can cause a loss of calcium homeostasis and the production of ROS. ROS are formed when XO degrades xanthine to produce uric acid; further, ROS originate from NOX activity, which is stimulated by muscle membrane depolarization (*Espinosa, Henríquez-Olguín & Jaimovich, 2016*). ROS also induce the NF-κB, which is actively involved in antioxidant regulation in response to oxidative stress. Activated NF-κB increases the release of pro-inflammatory mediators (*Rajakumar, Alexander & Oommen, 2013*). Eccentric exercises can also increase inflammatory mediators, such as TNFα, IL-1β, and IL-6 (*Paulsen et al., 2012*). Additionally, eccentric exercises also increase HSP-70 expression in skeletal muscles (*Mikkelsen et al., 2013*). Further, excessive expression of HSP-70 leads to the inhibition of NF-κB activation (*Tukaj, 2020*). HSP-70 protects against muscle damage and improves muscle recovery (*Senf, 2013*). HSP-70 also plays a role in preventing apoptosis by inhibiting caspase 3 activation (*Liu, 2006*). Muscle damage is indicated by increased fasting skeletal muscle Troponin I (TNNI2) levels during eccentric exercises (*Chapman et al., 2013b*). Levels of plasma TNNI2 are known to represent skeletal muscle damage more specifically than CK (*Chen et al., 2020*).

### NOX2 levels

This study revealed that an L-citrulline dose of 1,000 mg/kg can reduce NOX2 activation (*Tsuboi, Maeda & Hayashi, 2018*). This study also found that L-citrulline at doses of 250 and 500 mg/kg were not significantly different from those receiving no L-citrulline. This could be related to the fact that after entering the body, approximately 83% of L-citrulline reaches the kidneys for conversion to L-arginine (*Kaore & Kaore, 2014*). Despite conversion to other substances, constant supplementation of oral L-citrulline markedly increased plasma citrulline levels in a dose-dependent manner (*Schwedhelm et al., 2008*). In addition, previous study showed that high concentrations of exogenous L-citrulline are required to maintain maximal NOS activity (*Wileman et al., 2003*). NOS is used for the synthesis of NO which inhibits the activation of the NOX2 subunit. This may also explain the findings in this study, why only NOX2 levels in mice after momentary eccentric exercise given a dose of L-citrulline 1,000 mg/kgBW were lower when compared to mice without L-citrulline

administration. Furthermore, *Tsuboi, Maeda & Hayashi (2018)* also conducted research on the expression of NADPH oxidase subunits, such as p22phox, p47phox and p67phox using Western blot in rabbits as experimental animals. It was found that L-citrulline supplementation only significantly reduced the level of p67 [phox] protein (*Tsuboi, Maeda & Hayashi, 2018*). *Koju et al. (2019)* explained that the active NOX2 complex contains not only NOX2 and its partner subunit p22[phox] but also several regulatory subunits (p47[phox], p67[phox], Rac2) (*Koju et al., 2019*).

## NF-κB expression

The expression of NF-κB in post-eccentric exercise mice administered with L-citrulline at doses of 250, 500, and 1,000 mg/kg of BW was lower than without L-citrulline. NF-κB is one of the most important signaling pathways activated during eccentric exercise (*Jiménez-Jiménez et al., 2008*). NF-κB is useful for providing the first and fastest cellular stimulus–response for emergency conditions (*Rajakumar, Alexander & Oommen, 2013*). L-citrulline supplementation was reported to inhibit NF-κB activation significantly (*Darabi et al., 2019*; *Ba et al., 2022*). A significant decrease in NF-κB levels in the L-citrulline supplementation of 2 g/day groups after three months (*Darabi et al., 2019*). L-citrulline supplementation of 2 g/day was also possible in significantly reducing NF-κB levels in mice with iron overload-induced in the thymus (*Ba et al., 2022*). L-citrulline supplementation can reduce TLR4 gene expression, inhibiting NF-κB activation and TNF-α production (*Jegatheesan et al., 2016*). Another mechanism underlying the anti-inflammatory effects of L-citrulline is its ability to reduce oxidative stress. *Cai et al. (2016)* reported that L-citrulline supplementation increases superoxide dismutase (SOD) activity and reduces MDA levels (*Cai et al., 2016*). SOD can reduce the activation of extracellular signal-regulated protein kinase 1 and 2 (ERK1/2) signaling. ERK1/2 inhibition leads to the prevention of NF-κB activation and TNF-α production (*Perriotte-Olson et al., 2016*).

## HSP-70 expression

HSP-70 expression in mice administered with L-citrulline at 250, 500, and 1,000 mg/kg BW was lower than without L-citrulline. Previous studies have shown that physical exercise induces HSP-70 expression (*Heck et al., 2017*). HSP-70 protein levels increase after exercise in response to eccentric components associated with damage to human skeletal muscles (*Paulsen et al., 2012*), and the increased HSP-70 levels lead to a faster progression of muscle recovery (*Senf, 2013*). Exercise increases HSP-70 to protect against stress and suppress caspase 3 activity in mice (*Mikami et al., 2004*). These study results is consistent with those of *Petiz et al. (2017)*, who reported that antioxidant supplementation reduced the expression of anti-inflammatory IL-10 and HSP-70, which are important factors for exercise adaptation and prevention of tissue damage (*Petiz et al., 2017*). HSP-70 is regulated by HSF-1, activated by several stressors, such as heat and oxidative stress (*Liu, 2006*). A previous study showed that L-citrulline supplementation reduced HSF-1 levels in experimental chicks. L-citrulline supplementation significantly downregulated the expression of HSP-60; however, the downregulation was not significant for HSP-70 expression (*Uyanga et al., 2021*). In the present study, expression of HSP-70 in post-eccentric exercise mice administered with L-citrulline was lower than that in mice without

L-citrulline administration. Hence, L-citrulline administration possibly prevents muscle damage after acute eccentric exercise by strengthening endogenous antioxidants. These events generate a molecular balance, decreasing denatured protein levels and subsequently activating HSP-70.

Furthermore, HSP-70 has been recognized for its significant role in immune regulation and cell protection during exercise, contributing to the efficiency of regeneration and repair processes. It also holds diagnostic potential in the realm of sports science, allowing for the monitoring of the effects of exercise on skeletal muscle and the detection of muscle damage (*Krüger, Reichel & Zeilinger, 2019*). Notably, in the specific context of eccentric exercise following downhill running, it has been demonstrated to be more effective in eliciting the HSP-70 response in muscles compared to various other types of running (*Lollo et al., 2013*). This underscores the multifaceted role of HSP-70 in the exercise-induced responses and its potential application as a diagnostic marker in sports science research.

## IL-6 levels

IL-6 levels in mice administered L-citrulline at 250, 500, and 1,000 mg/kg BW were lower than those in mice not administered L-citrulline. IL-6 is an important myokine that exhibits pro-inflammatory or anti-inflammatory effects (*Nara & Watanabe, 2021*). Downhill exercise is known that increase IL-6 levels in the skeletal muscle (*Isanejad et al., 2015*). This study result is in line with *Fischer et al. (2004)* study, which found that antioxidant supplementation inhibited the release of IL-6 from contracting human skeletal muscles, which contracted approximately 50% less in the treatment group compared to controls (*Fischer et al., 2004*). Oxidative stress reduced by N-acetylcysteine decreases ERK1/2, p38, and extracellular NF-κB signaling proteins as well as reduces IL-6 formation (*Sigala et al., 2011*). L-citrulline improves blood flow and increases NO bioavailability, removing metabolites such as $H^+$ ions and free radicals from the muscle tissue and ultimately increasing antioxidant production. L-Citrulline supplementation before exercise can be an effective antioxidant agent that increases SOD, GPx, and CAT levels through its antioxidant properties. L-Citrulline has been shown to have a protective effect against reactive oxygen species (ROS) and oxidative stress, which may be related to its antioxidant capacity (*Allerton et al., 2018*). IL-6 levels in post-acute eccentric exercise mice administered L-citrulline were lower than those in mice without L-citrulline administration. They did not cross the basal threshold in the normal control group. This may be due to the effect of L-citrulline, which prevents muscle damage after acute eccentric exercise to strengthen the body's molecular response to oxidative stress.

## Caspase 3 levels

Caspase 3 levels in post-exercise eccentric mice administered L-citrulline at 250, 500, and 1,000 mg/kg BW were lower than those without L-citrulline. The dose of L-citrulline (250 mg/kg) showed the best results in reducing caspase 3 activation. Caspase 3 activity has been known to increase after eccentric exercise (*Townsend et al., 2018*). This study's results align with *Yu et al. (2023)* study, which stated that L-citrulline supplementation could inhibit caspase 3 activation (*Yu et al., 2023*). Thus, L-citrulline prevents muscle damage by

inhibiting caspase 3 activation. Increasing L-citrulline doses reduced the effect of caspase 3 activation inhibition in the skeletal muscle tissue of mice after acute eccentric exercise. Inhibition of caspase 3 activation may be due to increasing NO levels. This study did not examine NO levels, a limitation that should be used as a basis for further research.

### TNNI2 level

TNNI2 levels in post-eccentric exercise mice administered with l-citrulline (250 and 500 mg/kg BW) were lower than those without L-citrulline. L-citrulline (250 mg/kg) showed the best results in reducing plasma TNNI2 levels. The results of this study complement the findings of *Takeda et al. (2011)*, who reported that L-citrulline at a dose of 250 mg/kg BW for seven days effectively increased exercise performance in mice (*Takeda et al., 2011*). L-citrulline supplementation has also been shown to prevent muscle damage when administered before exercise; it stated that the L-citrulline supplementation using watermelon juice added with 3.45 g/500 mL L-citrulline could reduce the incidence of DOMS at 48 h after running a marathon (*Martínez-Sánchez et al., 2017*).

### Implication of study

L-citrulline may prevent skeletal muscle damage in mice after acute eccentric exercise through antioxidant effects as well as inflammatory and apoptotic pathways. Based on dose-related effects, it was found that L-citrulline doses of 250, 500, and 1,000 mg/kg significantly influenced the expression of NF-κB and HSP-70, as well as the levels of IL-6 and caspase 3. Meanwhile, only doses of 250 and 500 mg/kg had an impact on TNNI2 levels, and the 1,000 mg/kg dose affected NOX2 levels. L-citrulline supplementation are substantial for the sports field, especially may play a crucial role in reducing muscle damage and inflammation associated with eccentric exercise. These potential benefits could have a profound impact on athletes and individuals engaged in strenuous physical activities, offering a strategy to enhance exercise performance and expedite recovery. Mitigating muscle damage and inflammation may translate to improved exercise tolerance, reduced post-exercise soreness, and faster recovery, which can be especially advantageous for athletes and individuals seeking to optimize their athletic performance, minimize the risk of injury, and enhance their training regimens.

### Limitations of study

This study has some limitations. This study did not examine fast and slow muscle ratios and eNOS levels. Further, a simple randomization of mice was performed. It is also unknown whether the increase in NOX2 activity is related to muscle damage caused by L-citrulline supplementation. Inhibition of NOX2 activation by L-citrulline supplementation was only observed in the p67[phox] subunit; however, this study did not examine the NOX2 subunit. Lastly, the levels of NO were also not determined in this study. NO levels are shown to be associated with the activation of NF-κB, IL-6, and caspase 3 in the skeletal muscle tissue of mice after an acute eccentric exercise.

## CONCLUSIONS

This study has provided evidence suggesting that skeletal muscle damage (plasma TNNI2 levels) in mice after eccentric exercise was lower after administering L-citrulline at 250 and 500 mg/kg, followed by changes in markers of oxidative stress (NOX2) which were lower after administration of L-citrulline (1,000 mg/kg). Lower changes in cellular response markers for skeletal muscle damage (NF-κB, HSP-70, IL-6, and caspase 3) were observed after administration of L-citrulline (250, 500, and 1,000 mg/kg). Although the results of this study supported the role of L-citrulline in preventing skeletal muscle damage after acute eccentric exercise, the direct effect of L-citrulline in humans remains undetermined. Hence, further studies should be performed using this study as a guideline to understand the beneficial role of L-citrulline. L-citrulline doses may be a preventive therapy against skeletal muscle damage after an acute eccentric exercise. Hence, this study paves the way for future animal and clinical investigations to support the therapeutic translation of this supplement in patients with skeletal muscle damage.

### Funding
The authors received no funding for this work.

### Competing Interests
The authors declare there are no competing interests.

### Author Contributions
- Dhoni Akbar Ghozali conceived and designed the experiments, performed the experiments, analyzed the data, prepared figures and/or tables, authored or reviewed drafts of the article, and approved the final draft.
- Muchsin Doewes conceived and designed the experiments, authored or reviewed drafts of the article, and approved the final draft.
- Soetrisno Soetrisno conceived and designed the experiments, authored or reviewed drafts of the article, and approved the final draft.
- Dono Indarto conceived and designed the experiments, authored or reviewed drafts of the article, and approved the final draft.
- Muhana Fawwazy Ilyas analyzed the data, prepared figures and/or tables, authored or reviewed drafts of the article, and approved the final draft.

### Animal Ethics
The following information was supplied relating to ethical approvals (i.e., approving body and any reference numbers):

The Research Ethics Committee of Faculty of Medicine, Universitas Sebelas Maret, with protocol number of 01/02/09/2022/117

## Data Availability

The raw measurements are available in the Supplementary Files.

## Supplemental Information

Supplemental information for this article can be found online at http://dx.doi.org/10.7717/peerj.16684#supplemental-information.

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
