# Peer review of "Dose-response effect of L-citrulline on skeletal muscle damage after acute eccentric exercise: an in vivo study in mice"

_PeerJ, doi:10.7717/peerj.16684_

## Round 0.1 · original submission · Major Revisions

· Academic Editor

Major Revisions

Dear Authors

Both reviewers requested a major revision of the manuscript. These also include the introduction of a control and more information on the number of specimens in the experiment, the duration of the exercise etc.

You are also required to update the arrows on the micrograph or provide an explanation of what they intend to point to and include scale bars on all figures.

Reviewer 1 ·

Basic reporting

.

Experimental design

.

Validity of the findings

.

Additional comments

Comments to the authors
In this manuscript, the authors investigated through an in vivo approach, the effects of L-citrulline on skeletal muscle damage by screening the protein expression level of five markers (TNNI2, NOX2, IL-6, caspase 3, NF-κB and HSP 70).
The manuscript is well written, but to consider this manuscript for publication, authors should add some information as mentioned in the comments below.


GENERAL COMMENTS

1) Why the authors choose to investigate the levels of proteins in different matrices: TNNI2, NOX2, IL-6, and caspase 3 were determined by ELISA in plasma while NF-κB and HSP 70 expressions were determined by immunohistochemistry from tissues.
2) I was wondering if mTOR pathway roles mediated by citrulline may have a role in those protein expressions ?


ABSTRACT
1) Indicate the number of biological replicates for each group and not only the overall number of biological replicates
2) Better justify that you study evidences “L-citrulline may prevent skeletal muscle damage in mice after acute eccentric exercise through antioxidant effects as well as inflammatory and apoptotic pathways”. Based on you results it’s seems mostly to be a shortcut rather than a conclusion.


INTRODUCTION
3) Line 51 “One of the ergogenic supplements that may have a preventive effect on skeletal muscle damage is L- citrulline”. Cite references
4) Add the reference of the review that deals with the effect of amino acids including citrulline on proteome (Bourgoin-Voillard et al, 2016, DOI 10.1002/pmic.201500347)
5) Add also other important references that deal with the effects of L-citrulline on muscle
- Osowska, S., Duchemann, T., Walrand, S., Paillard, A. et al., Citrulline modulates muscle protein metabolism in old mal nourished rats. Am. J. Physiol. Endocrinol. Metab. 2006,
- Faure, C., Raynaud-Simon, A., Ferry, A., Dauge, V. et al., ´ Leucine and citrulline modulate muscle function in malnourished aged rats. Amino Acids 2012, 42, 1425–1433. 291, E582–E586.
- Le Plenier, S., Walrand, S., Noirt, R., Cynober, L., Moinard, ´ C., Effects of leucine and citrulline versus non-essential amino acids on muscle protein synthesis in fasted rat: a common activation pathway? Amino Acids 2012, 43, 1171– 1178.
6) What about the effects on protein biosynthesis and metabolism ?
7) Justify why the study of TNNI2 and HSP 70 is interesting


EXPERIMENTAL SECTION
1) Indicate the number of biological replicates for each group and not only the overall number of biological replicates
2) Line 95: “The mice were allowed to adapt to the Colombus Treadmill 96 (Columbus Instruments, Columbus, OH, USA) for 5 minutes, and subsequently, downhill 97 running was conducted at a speed of 30 cm/s for 18 min at a -15° declination angle.” Complete the sentence by indicate the frequency of running. Running was how long time after the absorbance and/or feeding of citrulline?
3) Add a control to check muscle damage evolution


RESULTS
1) Figure 1: you should indicate on the line the significance of P-value: * <0.05; **<0.01; ***<0.005…
2) Line 166: the authors wrote “Expression levels of HSP-70 in mice gastrocnemius muscle tissue are a marker of apoptosis…”, but you have other better marker of apoptosis if you want to determine the apoptosis.

DISCUSSION
1) Specificity issue of Hsp70 that is involve in many biological processes
2) Line 214: NOX2 levels. The authors highlighted the conversion of citrulline to explain the low effect at low concentration of exposure. What about possible antagonist effects and activation of different pathways?

Reviewer 2 ·

Basic reporting

No comment

Experimental design

No comment

Validity of the findings

No comment

Additional comments

The article by Ghozali and co-authors investigated the dose-response effect of L-citrulline supplementation on skeletal muscle damage after eccentric running exercise in mice. L-citrulline intake reduced eccentric exercise-induced muscle damage by evaluating plasma TNNI2 and muscle NOX2, NF-κB, HSP 70, IL-6, and caspase 3. Although this paper demonstrates the effectiveness of L-citrulline supplementation, the following details need to be revised.

Major
It is difficult to understand about interpretation of immunohistochemical staining results. In Figure 2, it looks that the arrows are not used in a consistent manner. Though some arrows indicate nuclei, other arrows indicate muscle fiber. Is this considering the localization of NF-κB and HSP70 in muscle fibers? If so, authors should discuss about this point in the main text.

The authors use ‘dose-response effect’ in the article title, so please argue this point in discussion section.

Please discuss specifically what the authors suppose the benefits of taking L-citrulline are to the sports field. For example, it may help to suppress delayed onset muscle soreness and improve the performance during endurance exercise.

Minor
L53&54 Describe the full spelling of NOS and NO.
L58&59 Describe the full spelling of ROS and NOX2.
L91 Describe in detail how did the authors fed citrulline to the mice.
L114 Describe the details about ELISA analysis. I could not understand the analysis conditions from skeletal muscle and plasma.
L121 Describe the details about immunohistochemical staining procedure and data analysis. Provide more information about this method. For example, preparation of skeletal muscle sections, name of reagents, diluted concentration of antibodies etc.
L152 Please correct the comma to period (2,33 ±0.70 ng/ml)
L152-153 The value of T2 and T3 are different from Table 1.
L174 Please correct IL-5 to IL-6.
L181 The data of T2 group is not same as Table 1.
L256&257&276 Please capitalize the l to L.
L285 Please correct TNII2.
Unify the expression of caspase 3 throughout the article. There are some patterns of caspase 3. (caspase 3, caspase-3, caspase three)
Table 1 and Figure 1 The data of NF-κB and HSP70 are not same between table and figure.
Figure 2 Put the scale bars in each figure.

---

## Round 0.2 · Minor Revisions

· Academic Editor

Minor Revisions

Dear Editor

The reviewer has now accepted acceptance of your manuscript provided that you attend to the following.
he manuscript has been much improved about previously pointed out. Therefore, it can be concluded that this paper is acceptable. However, the following points need to be corrected.

・It is not enough to unify the expression of caspase-3.
・The expressions of NF-kB dose not unify in Figure 2 legend.

Reviewer 2 ·

Basic reporting

The manuscript has been much improved about previously pointed out. Therefore, it can be concluded that this paper is acceptable. However, the following points need to be corrected.

・It is not enough to unify the expression of caspase-3.
・The expressions of NF-kB dose not unify in Figure 2 legend.

Experimental design

no comment

Validity of the findings

no comment

---

## Round 0.3 · accepted · Accept

· Academic Editor

Accept

All comments have been addressed.